# Global burden of NAFLD 1990–2021 and projections to 2035: Results from the Global Burden of Disease study 2021

Yuqin Mao[1☯], Jiqing Du[2☯], Baoguo Li[3,4☯], Jiong Wang[1], Shaoyan Xuan[1], Shu Yang[1], Zhihua Tang[1]*, Minxiu Wang[1]*

1 Department of Pharmacy, Shaoxing People's Hospital, School of Medicine, Shaoxing University, Shaoxing, Zhejiang, People's Republic of China, 2 School of Life and Health Technology, Dongguan University of Technology, Dongguan, China, 3 Tianjian Laboratory of Advanced Biomedical Sciences, Academy of Medical sciences, Zhengzhou University, Zhengzhou, Henan, China, 4 Innovation Center of Basic Research for Metabolic-Associated Fatty Liver Disease, Ministry of Education of China, Zhengzhou, China

☯ These authors contributed equally to this work.
* wmxstu123@163.com (MW); sxtzh@163.com (ZT)

## Abstract

### Background

Non-alcoholic fatty liver disease (NAFLD) is the most common chronic liver disease. The global burden of NAFLD is increasing. This study used the Global Burden of Disease (GBD) 2021 study data to assess the burden and development trends of NAFLD from 1990 to 2021.

### Methods

The incidence, prevalence, death and disability-adjusted life years (DALYs) rates of NAFLD in geographic populations worldwide from 1990 to 2021 were extracted from the GBD 2021 study data. The global temporal trend of NAFLD from 1990 to 2021 was evaluated using estimated annual percentage change and age-standardized rate. The Bayesian age-period-cohort model was used to predict NAFLD burden future trends to 2035.

### Results

The global age standardized incidence rate (ASIR) of NAFLD among the all-age population increased by 25% and the age standardized prevalence rate (ASPR) increased by 24%. The age standardized mortality rate (ASMR) and age standardized DALYs rate (ASDR) were relatively stable. Countries with middle socio-demographic index (SDI) had the highest ASIR and ASPR from 1990 to 2021, high-middle SDI and high SDI had the lowest ASMR and ASDR. North Africa and Middle East had the highest ASIR and ASPR in 2021, ASIR increased at the greatest

**Data availability statement:** The datasets for this article are available from the Global Health Data Exchange query tool (http://ghdx.healthda-ta.org/gbd-results-tool).

**Funding:** This study was supported by the Shanghai Medical Innovation and Development Foundation, the Yangtze River Delta pharmaceutical high-quality development research promotion program (SMIDF-142-3 to M.W.).

**Competing interests:** No authors have competing interests.

**Abbreviations:** NAFLD: non-alcoholic fatty liver disease; GBD: Global Burden of Disease; ASR: Age-standardized rate; ASIR: Age-standardized incidence rate; ASPR: Age-standardized prevalence rate; ASMR: age-standardized mortality rate; ASDR: Age-standardized DALYs rate; DALYs: Disability-adjusted life years; EAPC: Estimated annual percentage change; SDI: Socio-demographic index; UI: Uncertainty interval; CI: confidence interval; BAPC: Bayesian age-period-cohort.

rate in East Asia and Western Europe, ASPR increased at the greatest rate in Western Europe. In 2021, the highest number of incidence cases and incidence rates were in 20–24 years group, reflecting a tendency towards a younger onset of NAFLD.

## Conclusions

The global burden of NAFLD has risen steadily from 1990 to 2021, and projections to 2035, placing enormous pressure on society. It is necessary to implement measures targeting risk factors such as high fasting plasma glucose and tobacco, including improving lifestyle, adjusting diet, and applying drug intervention.

## Introduction

Non-alcoholic fatty liver disease (NAFLD), recently rebranded as metabolic-associated fatty liver disease (MAFLD) [1], is one of the most common liver diseases in the world [2]. According to recent systematic reviews and meta-analyses, the global prevalence of NAFLD has increased from 25.3% in 1990–2006 to 38.0% in 2016–2019 [3]. NAFLD comprises a spectrum of disorders typified by an excessive deposition of fat in the liver, excluding alcoholic intake and other identified liver-damaging agents [4]. Characterized by an excessive hepatic fat accumulation and accompanied by insulin resistance, it is histologically defined by the presence of steatosis affecting more than 5% of hepatocytes [5]. NAFLD encompasses an excessive accumulation of lipids within the liver, leading to lipotoxicity, with potential progression to metabolic-associated steatohepatitis (NASH), liver fibrosis, and hepatocellular carcinoma. It stands as one of the principal causes of hepatocellular carcinoma (HCC) and liver transplantation [6].

Extensive epidemiological research has revealed that the pathophysiological ramifications of NAFLD extend beyond the confines of the liver [7]. Clinical evidence robustly establishes NAFLD as an independent risk factor for the prevalence and onset of cardiovascular disease (CVD) [8–10], chronic kidney disease (CKD) [11], and type 2 diabetes mellitus (T2DM) [12,13], thereby elevating the susceptibility to significant extrahepatic chronic illnesses, including CVD, T2DM, and CKD. T2DM and obesity are closely associated with a higher risk of NAFLD. It is estimated that globally, 55–70% of patients with type 2 diabetes have NAFLD, 30–60% have NASH, and 12–20% have clinically significant fibrosis (≥ stage 2) [14]. With the rising prevalence of obesity and type 2 diabetes, as well as population aging, the burden of NAFLD is expected to increase in the next decade.

A significant racial disparity is evident in the prevalence of NAFLD, with the highest rates documented in individuals of Latin descent, the lowest in African Americans, and intermediate rates in Caucasians and Asians [15]. Analogous to Western countries, the incidence of NAFLD in Asian countries is on the rise, fueled by an increase in cases of obesity, type 2 diabetes, and other components of the metabolic syndrome [16,17]. As the global obesity epidemic exacerbates metabolic conditions, the clinical and economic toll of NAFLD is set to become staggering [15].

Although NAFLD has reached a pandemic proportion globally, no specific pharmacological treatments have been approved for its management [18]. While anti-diabetic, lipid-lowering, and natural bile acid therapies have been utilized in the treatment of NAFLD, they are accompanied by certain drawbacks [19]. Consequently, the analysis of up-to-date statistical data is crucial for the effective prevention, control, and management of NAFLD. In the present study, we conducted a retrospective examination of the worldwide NAFLD disease burden utilizing data from the Global Burden of Disease (GBD) 2021, thereby providing contemporary estimates for the advancement of ongoing epidemiological research on NAFLD.

## Materials and methods

### Study data

This study used data on the incidence, prevalence, deaths and DALYs attributable to NAFLD and the relevant risk factors from 1990 to 2021 obtained from the Global Health Data Exchange (GHDx) query tool (http://ghdx.healthdata.org/gbd-results-tool), based on gender, age, region, and country. The 204 countries and territories were categorized into five levels based on their Socio-Demographic Index (SDI), which includes high, high middle, middle, low middle, and low SDI. SDI was calculated using several social factors, including the fertility rate of the population aged <25 years, the education level of the population aged >15 years, and per capita income [20]. According to geographical contiguity, the world was divided geographically into 21 different GBD regions, such as Western Europe and East Asia. The data obtained from the GHDx Query Tool did not require informed patient consent and was publicly available.

### Statistical analysis

The trends for NAFLD incidence, prevalence, and mortality rates were evaluated by calculating the age-standardized incidence rate (ASIR), age-standardized prevalence rate (ASPR), age-standardized mortality rate (ASMR), age-standardized DALYs rate (ASDR), and their respective estimated annual percentage change (EAPC). DALYs were computed as the sum of the years lived with disability and the years of life lost. According to the age group construction of the standard population, the ASR (per100, 000 population) were calculated using the following formula:

$$ASR = \frac{\sum_{i=1}^{A} a_i w_i}{\sum_{i=1}^{A} a_i} \times 100,000,$$

($a_i$ refers to the incidence of the $i^{th}$ age group. $w_i$ denotes the number of persons (or weight) in the same age subgroup $i$ of the assigned reference standard population)

EAPC is used to estimate the trends of the ASRs, and it quantitatively calculates the average annual rate of change of ASR for a specified period [21], which follows this formula, i.e., $y = \alpha + \beta x + \varepsilon$, where $y = \ln(ASR)$, and $x =$ the calendar year. The EAPC calculation formula, $100 \times (\exp(\beta) - 1)$, and its 95% confidence intervals (CI) can also be calculated from the linear regression model.

The Bayesian age-period-cohort (BAPC) model can achieve more reasonable predictions to the global burden trends. Based on the assumption that the effects of age, period, and cohort were similar in temporal proximity, Bayesian inference in the BAPC model utilized a second-order stochastic excursion to smooth the prior three aforementioned values and forecast the posterior rates. An integrated nested Laplacian approximation was used with this BAPC model to approximate marginal posterior distributions, avoiding some of the mixing and convergence problems introduced by the Markov Chain Monte Carlo sampling technique traditionally used for Bayesian methods [22]. To ensure smoothing, BAPC models assume independent mean-zero normal distributions on the second differences of all effects. Specifically, the BAPC model assumes prior distribution of the age effect as follows:

$$f(\alpha|k_\alpha) \propto k_\alpha^{\frac{t-2}{2}} \exp\left\{-\frac{k_\alpha}{2}\sum_{i=3}^{I}[(\alpha_i - \alpha_{i-1}) - (\alpha_{i-1} - \alpha_{i-2})]^2\right\},$$

Here, we add an independent random effect $\delta_{a,p+t} \sim N\left(0, k_\delta^{-1}\right)$ to adjust for overdispersion [1]. Considering the smoothing assumption, the BAPC models assume prior distribution of the period effect as follows:

$$\beta_{p+t}|\beta_1, \ldots, \beta_p, k_\beta \sim N((1+t)\beta_p - t\beta_{p-1}, k_\beta^{-1}(1 + 2^2 + \cdots + t^2)),$$

The summary estimates (mean, standard deviation, 2.5% quantile, median and 97.5% quantile) of all variance parameters in the BAPC models can be obtained [23].

All data analyses were performed using the open-source software R (version 4.3.3). Statistical significance was set at $P < 0.05$.

## Results

### NAFLD burden at global level

Globally, the incident cases of NAFLD increased by 94% between 1990 and 2021, from 24.86 million to 48.35 million. The age standardized incidence rate (ASIR) of NAFLD among the all-age population increased by 25%, from 475.54 per 100 000 population in 1990 to 593.28 per 100 000 population in 2021, with the estimated annual percentage change (EAPC) was 0.73 (Table 1).

Similarly, the prevalent cases of NAFLD increased by 125% between 1990 and 2021, from 0.56 billion to 1.27 billion. The age standardized prevalence rate (ASPR) of NAFLD among this age group increased by 24%, from 12085.09 per 100 000 population in 1990 to 15018.07 per 100 000 population in 2021, with the EAPC was 0.73 (Table 2).

The death cases and DALYs of NAFLD also increased between 1990 and 2021. Death cases increased from 59.54 thousand to 138.33 thousand, with DALYs increased from 1689.25 thousand to 3667.27 thousand. However, the age standardized mortality rate (ASMR) and age standardized DALYs rate (ASDR) had no obvious changes. ASMR changed from 1.53 per 100 000 population in 1990 to 1.62 per 100 000 population in 2021. ASDR changed from 40.20 per 100 000 population in 1990 to 42.40 per 100 000 population in 2021. (Tables 3 and 4).

### NAFLD burden at national level

At the national level in 2021, Egypt had the highest ASIR of NAFLD (1188.57 per 100 000 population), followed by Kuwait (1174.18 per 100 000 population), Iran (1168.68 per 100 000 population). Finland had the lowest ASIR of NAFLD (310.17 per 100 000 population), followed by Canada (311.21 per 100 000 population). Kuwait and Egypt also had the highest ASPR of NAFLD (32312.17 per 100 000 population, 31668.80 per 100 000 population). Japan and Finland had the lowest ASPR of NAFLD (8133.47 per 100 000 population, 8358.51 per 100 000 population) (Fig 1A, 1B, S1 Table). In 2021, Egypt and Mongolia had the highest ASMR of NAFLD (9.44 per 100 000 population, 8.64 per 100 000 population). Mexico and Egypt had the highest ASDR of NAFLD (201.86 per 100 000 population, 196.55 per 100 000 population) (Fig 1C, 1D, S1 Table).

At the national level from 1990 to 2019, both ASIR and ASPR showed an increase in all countries and territories. This means that the burden of NAFLD continues to increase. ASIR increased at the greatest rate in Equatorial Guinea (EAPC: 1.19), followed by Oman (EAPC: 0.99). ASPR increased at the greatest rate in Italy (EAPC: 1.21), followed by Oman (EAPC: 1.03) (Fig 2A, 2B, S2 Table). ASMR and ASDR had increased or decreased in various countries around the global. Russian Federation had the highest ASMR (EAPC: 4.25) and ASDR (EAPC: 4.82) increased rates (Fig 2C, 2D, S2 Table).

### NAFLD burden at regional level

At the regional level, ASIR and ASPR presented inverted-U shape relationship with SDI (Fig 4A, 4B). Middle SDI had the highest ASIR and ASPR from 1990 to 2021. High SDI had the lowest ASIR and ASPR from 1990 to 2021 (Tables 1 and 2,

**Table 1. Incident cases and ASIR of NAFLD in 1990 and 2021, and temporal trends.**

| | 1990 | | 2021 | | 1990-2021 EAPC of ASPR (95%CI) |
|---|---|---|---|---|---|
| | Incident cases, No. × 10³ | ASIR per 100,000 (95% UI) | Incident cases, No. × 10³ | ASIR per 100,000 (95% UI) | |
| Global | 24856.16(22579.70-27333.11) | 475.54(432.59-518.19) | 48353.27(44229.14-52358.02) | 593.28(542.72-643.70) | 0.73(0.69-0.77) |
| **Socio-Demographic Index** | | | | | |
| Low SDI | 1925.96(1735.02-2127.91) | 483.42(440.22-527.63) | 5396.91(4857.49-5987.67) | 553.74(503.53-605.08) | 0.44(0.41-0.47) |
| Low-middle SDI | 5235.75(4734.02-5777.13) | 520.74(474.21-566.64) | 12064.46(10980.81-13233.27) | 623.22(569.61-676.22) | 0.59(0.55-0.62) |
| Middle SDI | 9002.61(8141.91-9945.64) | 533.69(485.35-580.65) | 17017.31(15552.70-18416.00) | 656.97(602.08-712.86) | 0.68(0.64-0.72) |
| High-middle SDI | 5404.82(4893.94-5906.25) | 478.36(435.28-522.10) | 8509.33(7857.78-9204.00) | 611.29(557.97-665.56) | 0.80(0.71-0.90) |
| High SDI | 3262.72(2977.82-3550.60) | 342.72(312.23-373.46) | 5326.14(4910.09-5729.42) | 450.03(412.20-488.43) | 1.00(0.95-1.05) |
| **Region** | | | | | |
| Andean Latin America | 171.83(155.69-189.72) | 494.93(453.46-540.75) | 399.22(366.92-433.22) | 582.61(536.86-630.67) | 0.55(0.54-0.56) |
| Australasia | 62.86(56.93-68.90) | 288.08(261.95-315.06) | 117.96(108.10-127.03) | 358.84(328.05-390.77) | 0.72(0.68-0.76) |
| Caribbean | 178.95(162.84-196.25) | 513.74(471.20-559.15) | 287.88(264.11-310.03) | 576.80(528.70-623.41) | 0.42(0.41-0.44) |
| Central Asia | 347.73(317.65-383.98) | 530.14(485.07-576.45) | 583.56(530.90-636.27) | 612.97(560.97-665.27) | 0.52(0.47-0.56) |
| Central Europe | 548.34(502.23-593.27) | 419.01(384.12-454.71) | 569.43(525.60-613.24) | 471.53(433.07-510.90) | 0.40(0.38-0.43) |
| Central Latin America | 881.46(800.89-971.67) | 581.20(531.70-632.78) | 1786.20(1641.45-1939.01) | 669.44(615.97-726.19) | 0.48(0.47-0.49) |
| Central Sub-Saharan Africa | 193.05(173.26-215.53) | 434.61(395.27-476.05) | 549.88(493.16-610.87) | 469.08(426.91-513.72) | 0.22(0.19-0.26) |
| East Asia | 6397.50(5758.12-7055.94) | 495.87(449.65-540.96) | 9905.42(9094.62-10735.34) | 620.37(565.23-675.90) | 0.73(0.57-0.90) |
| Eastern Europe | 1003.38(918.19-1086.54) | 418.62(384.11-454.77) | 1035.05(952.60-1117.15) | 471.81(433.23-511.73) | 0.41(0.37-0.44) |
| Eastern Sub-Saharan Africa | 702.06(625.66-783.55) | 468.64(425.87-512.55) | 1953.11(1744.60-2184.44) | 522.51(475.30-570.84) | 0.36(0.34-0.39) |
| High-income Asia Pacific | 591.33(539.68-643.48) | 310.56(282.75-338.69) | 728.25(664.40-786.98) | 348.79(319.19-378.61) | 0.53(0.45-0.61) |
| High-income North America | 974.20(886.65-1063.60) | 321.77(293.45-350.95) | 1635.65(1495.03-1770.49) | 391.24(357.04-425.57) | 0.70(0.68-0.73) |
| North Africa and Middle East | 2621.74(2382.53-2885.70) | 849.02(777.99-920.82) | 6578.95(6067.96-7080.11) | 1037.64(963.01-1109.65) | 0.70(0.67-0.74) |
| Oceania | 31.40(28.27-34.94) | 542.51(493.89-593.47) | 76.65(69.53-84.51) | 588.57(538.68-640.66) | 0.27(0.23-0.30) |
| South Asia | 4362.69(3921.92-4818.17) | 464.89(421.18-507.16) | 10765.35(9758.06-11817.45) | 564.19(513.27-615.45) | 0.61(0.55-0.67) |
| Southeast Asia | 2347.04(2125.15-2601.00) | 541.66(491.92-590.09) | 4606.20(4201.35-5033.59) | 622.70(569.31-678.69) | 0.48(0.46-0.50) |
| Southern Latin America | 154.12(140.38-168.85) | 313.83(285.65-343.42) | 283.02(258.97-307.04) | 386.42(352.60-420.05) | 0.68(0.64-0.72) |
| Southern Sub-Saharan Africa | 273.21(245.84-304.04) | 570.40(520.52-622.36) | 532.71(484.10-584.24) | 651.93(597.18-709.27) | 0.47(0.46-0.48) |
| Tropical Latin America | 806.67(732.24-888.59) | 557.12(509.05-604.17) | 1584.12(1456.86-1706.17) | 648.83(596.36-700.52) | 0.55(0.53-0.58) |
| Western Europe | 1387.07(1270.98-1508.21) | 324.92(296.52-354.41) | 1871.64(1719.78-2008.90) | 401.69(368.55-435.70) | 0.73(0.68-0.78) |
| Western Sub-Saharan Africa | 818.84(737.07-907.48) | 521.28(474.89-569.13) | 2501.86(2243.38-2767.10) | 597.04(544.27-651.74) | 0.44(0.43-0.45) |

*ASIR* Age-standardized incidence rate, *EAPC* Estimated annual percentage change, *CI* Confidence interval, *SDI* Socio-demographic index, *UI* Uncertainty interval.

**Table 2. Prevalent cases and ASPR of NAFLD in 1990 and 2021, and temporal trends.**

| | 1990 | | 2021 | | 1990-2021 EAPC of ASPR (95%CI) |
|---|---|---|---|---|---|
| | Prevalent cases, No. ×10³ | ASPR per 100,000 (95% UI) | Prevalent cases, No. ×10³ | ASPR per 100,000 (95% UI) | |
| **Global** | 564432.13(516524.82-618101.25) | 12085.09(11058.41-13184.29) | 1267868.00(1157934.07-1380435.42) | 15018.07(13756.47-16361.44) | 0.73(0.67-0.79) |
| **Socio-Demographic Index** | | | | | |
| **Low SDI** | 40088.81(36552.30-44255.67) | 12421.48(11351.33-13593.54) | 107435.46(98307.36-118879.40) | 13889.22(12685.44-15166.26) | 0.35(0.31-0.39) |
| **Low-middle SDI** | 113447.13(103912.27-124978.28) | 13578.98(12385.81-14820.28) | 275895.44(251999.52-301392.88) | 15748.70(14418.61-17126.74) | 0.49(0.44-0.53) |
| **Middle SDI** | 192822.30(176670.36-212169.96) | 13847.03(12679.85-15124.73) | 453616.04(414597.79-494648.03) | 16589.92(15168.29-18069.08) | 0.61(0.54-0.69) |
| **High-middle SDI** | 131193.56(120116.96-143511.51) | 12230.41(11204.06-13334.14) | 260953.82(239296.77-283534.84) | 15471.06(14155.43-16878.90) | 0.78(0.67-0.89) |
| **High SDI** | 86299.85(79056.14-94042.90) | 8520.08(7809.16-9274.20) | 168913.42(154930.23-182979.40) | 11543.68(10585.57-12529.08) | 1.09(1.04-1.14) |
| **Region** | | | | | |
| **Andean Latin America** | 3550.47(3245.58-3891.05) | 12946.17(11857.20-14128.82) | 9737.66(8884.04-10649.71) | 14984.76(13708.36-16361.47) | 0.53(0.51-0.55) |
| **Australasia** | 1656.06(1512.94-1804.38) | 7376.03(6738.42-8024.72) | 3778.62(3457.86-4124.39) | 9468.25(8665.51-10349.36) | 0.83(0.80-0.87) |
| **Caribbean** | 4233.44(3871.41-4631.11) | 14073.50(12862.57-15354.69) | 8111.96(7440.86-8804.73) | 15650.72(14340.15-16986.20) | 0.38(0.37-0.40) |
| **Central Asia** | 7937.27(7270.98-8700.39) | 14118.98(12962.74-15421.66) | 15204.17(13885.52-16673.76) | 16120.05(14735.29-17604.48) | 0.45(0.41-0.48) |
| **Central Europe** | 15977.95(14560.92-17364.34) | 11366.94(10414.56-12359.43) | 20606.02(18822.10-22372.11) | 12731.47(11618.63-13852.64) | 0.39(0.37-0.41) |
| **Central Latin America** | 17471.42(16058.56-19196.19) | 14960.34(13710.10-16309.99) | 44693.57(40900.17-48782.24) | 16983.98(15536.52-18533.59) | 0.44(0.43-0.45) |
| **Central Sub-Saharan Africa** | 3761.67(3403.64-4187.89) | 10904.86(9951.55-12009.31) | 10850.62(9833.00-11986.28) | 11870.63(10844.86-12943.36) | 0.28(0.25-0.32) |
| **East Asia** | 142094.26(129881.12-156713.75) | 12789.93(11676.75-14020.27) | 301408.39(274406.34-328824.04) | 15596.18(14262.43-16999.34) | 0.67(0.45-0.90) |
| **Eastern Europe** | 28935.07(26460.11-31524.21) | 11080.88(10133.33-12074.69) | 34696.29(31695.85-37763.37) | 12293.91(11254.47-13359.17) | 0.32(0.31-0.33) |
| **Eastern Sub-Saharan Africa** | 13521.29(12352.29-14926.69) | 11774.95(10738.60-12873.14) | 37304.08(34041.06-41466.67) | 13162.12(12037.13-14400.23) | 0.35(0.33-0.37) |
| **High-income Asia Pacific** | 15202.61(13914.42-16576.61) | 7690.99(7025.96-8379.61) | 24694.24(22636.60-26784.25) | 8885.73(8148.40-9666.66) | 0.58(0.51-0.65) |
| **High-income North America** | 25568.85(23386.40-27948.79) | 7946.82(7253.21-8661.46) | 48995.59(44673.05-53423.29) | 10056.02(9187.35-10925.56) | 0.89(0.85-0.93) |
| **North Africa and Middle East** | 52713.32(48290.12-57887.15) | 21902.49(20094.34-23849.75) | 164312.59(151441.88-179050.65) | 27686.69(25586.92-29914.62) | 0.82(0.77-0.87) |
| **Oceania** | 605.23(551.74-670.03) | 13706.50(12511.17-15026.99) | 1625.69(1483.82-1796.52) | 15182.69(13936.16-16584.71) | 0.35(0.32-0.39) |
| **South Asia** | 99245.06(90560.95-109619.91) | 12361.12(11282.78-13532.19) | 249790.70(227865.12-273237.52) | 14158.33(12940.87-15445.08) | 0.43(0.34-0.51) |
| **Southeast Asia** | 49379.73(45181.58-54280.55) | 13984.92(12785.15-15266.89) | 115103.79(104841.85-125940.61) | 15691.67(14308.28-17127.24) | 0.39(0.38-0.41) |

*(Continued)*

**Table 2.** (Continued)

| | 1990 | | 2021 | | 1990-2021 EAPC of ASPR (95%CI) |
|---|---|---|---|---|---|
| | **Prevalent cases, No. × 10³** | **ASPR per 100,000 (95% UI)** | **Prevalent cases, No. × 10³** | **ASPR per 100,000 (95% UI)** | |
| **Southern Latin America** | 3801.55(3476.77-4182.35) | 7995.98(7321.85-8783.11) | 8080.74(7374.98-8823.03) | 10292.48(9394.62-11265.42) | 0.84(0.80-0.89) |
| **Southern Sub-Saharan Africa** | 5251.57(4811.69-5773.20) | 13932.66(12701.43-15196.92) | 11781.79(10763.17-12907.45) | 15937.24(14572.60-17388.03) | 0.43(0.41-0.46) |
| **Tropical Latin America** | 18156.84(16647.04-20103.25) | 15001.90(13772.66-16413.10) | 42870.51(39161.07-46852.27) | 16662.75(15244.95-18205.46) | 0.39(0.37-0.40) |
| **Western Europe** | 38629.11(35378.10-41988.28) | 8144.98(7490.53-8873.81) | 66258.94(60940.04-71561.63) | 10841.79(9939.03-11801.96) | 0.98(0.93-1.03) |
| **Western Sub-Saharan Africa** | 16739.35(15345.72-18454.59) | 13317.94(12164.76-14554.97) | 47962.04(43856.19-53002.94) | 14936.85(13659.28-16347.60) | 0.35(0.33-0.36) |

*ASIR* Age-standardized prevalence rate, *EAPC* Estimated annual percentage change, *CI* Confidence interval, *SDI* Socio-demographic index, *UI* Uncertainty interval.

Fig 3). In 2021, Low-middle SDI had the highest ASMR and ASDR, with High-middle SDI had the lowest ASMR and ASDR (Tables 3 and 4, Fig 3). Low-middle SDI and Middle SDI showed an increase in ASMR and ASDR from 1990 to 2021 (Fig 3, S3 Table).

From 1990 to 2021, all 21 regions showed a sustained rise in ASIR and ASPR (Fig 4A, 4B). North Africa and Middle East had the highest ASIR (1037.64 per 100 000 population) and ASPR (27686.69 per 100 000 population) in 2021 (Tables 1 and 2), higher than that of the expected level based on SDI from 1990 to 2021 (Fig 4A, 4B, S4 Table). ASIR increased at the greatest rate in East Asia (EAPC: 0.73) and Western Europe (EAPC: 0.73). ASPR increased at the greatest rate in Western Europe (EAPC: 0.98), followed by High-income North America (EAPC: 0.89) (Tables 1 and 2).

In 2021, Andean Latin America and Central Latin America had the highest ASMR (5.89 per 100 000 population, 5.00 per 100 000 population) and ASDR (142.16 per 100 000 population, 138.84 per 100 000 population) (Tables 3,4), higher than that of the expected level based on SDI from 1990 to 2021 (Fig 4C, 4D, S4 Table). Eastern Europe had the highest ASMR (EAPC: 3.58) and ASDR (EAPC: 4.09) increased rates (Tables 3,4).

## Age and sex patterns

In 2021, the highest number of incidence cases and incidence rates were in 20–24 years group, and these gradually declined with age (Fig 5A, S5 Table). The incidence of NAFLD in young people was higher than that in middle-aged and older people, reflecting a tendency towards a younger onset of NAFLD. The number of incidence cases and incidence rates were apparently higher in males than in females before the age of 45, which may mean that younger men are more likely to develop NAFLD (Fig 5A).

In 2021, the number of prevalence cases presented inverted-U shape relationship with age, the highest number of prevalence cases was in 45–49 years group. The prevalence rates of NAFLD gradually increased with age until it peaked in 75–79 years group (Fig 5B).

ASMR and ASDR were observed to exhibit a progressive increase with advancing age. The number of death cases and DALYs presented inverted-U shape relationship with age. The highest number of death cases was in 65–69 years group, with the highest number of DALYs was in 55–59 years group (Fig 5C, 5D).

**Table 3. Death cases and ASMR of NAFLD in 1990 and 2021, and temporal trends.**

| | 1990 | | 2021 | | 1990-2021 EAPC of ASMR (95%CI) |
|---|---|---|---|---|---|
| | Death cases, No.×10³ | ASMR per 100,000 (95% UI) | Death cases, No.×10³ | ASMR per 100,000 (95% UI) | |
| **Global** | 59.54(45.67-76.30) | 1.53(1.17-1.97) | 138.33(108.29-173.90) | 1.62(1.27-2.02) | 0.19(0.14-0.24) |
| **Socio-Demographic Index** | | | | | |
| **Low SDI** | 4.03(2.91-5.55) | 1.89(1.34-2.67) | 8.41(6.56-10.87) | 1.74(1.34-2.23) | −0.32(−0.36--0.28) |
| **Low-middle SDI** | 9.65(6.96-13.47) | 1.69(1.20-2.43) | 26.19(19.44-33.98) | 1.87(1.39-2.41) | 0.37(0.35-0.39) |
| **Middle SDI** | 16.08(12.37-20.44) | 1.63(1.25-2.11) | 46.55(36.62-58.52) | 1.78(1.39-2.21) | 0.36(0.33-0.40) |
| **High-middle SDI** | 14.18(10.78-18.10) | 1.47(1.12-1.86) | 27.01(20.90-34.20) | 1.40(1.09-1.76) | −0.14(−0.30-0.01) |
| **High SDI** | 15.52(11.78-20.04) | 1.43(1.09-1.84) | 30.01(23.14-37.48) | 1.48(1.15-1.85) | 0.13(0.05-0.22) |
| **Region** | | | | | |
| **Andean Latin America** | 0.97(0.68-1.33) | 4.73(3.32-6.45) | 3.44(2.36-4.73) | 5.89(4.03-8.07) | 0.71(0.62-0.80) |
| **Australasia** | 0.18(0.14-0.23) | 0.78(0.60-1.01) | 0.66(0.52-0.81) | 1.27(1.01-1.55) | 1.77(1.63-1.91) |
| **Caribbean** | 0.72(0.52-0.96) | 2.82(2.04-3.72) | 1.60(1.13-2.20) | 2.97(2.11-4.07) | 0.13(−0.13-0.38) |
| **Central Asia** | 1.01(0.75-1.33) | 2.17(1.60-2.86) | 2.75(1.99-3.73) | 3.41(2.47-4.59) | 1.66(1.42-1.89) |
| **Central Europe** | 2.02(1.50-2.69) | 1.37(1.02-1.79) | 3.50(2.57-4.72) | 1.69(1.24-2.27) | 0.35(0.19-0.51) |
| **Central Latin America** | 3.59(2.59-4.76) | 4.21(3.04-5.56) | 12.63(9.45-16.29) | 5.00(3.74-6.45) | 0.56(0.48-0.63) |
| **Central Sub-Saharan Africa** | 0.40(0.26-0.63) | 1.76(1.14-2.85) | 0.91(0.59-1.38) | 1.61(1.04-2.50) | −0.41(−0.50--0.33) |
| **East Asia** | 8.69(6.74-10.95) | 1.04(0.80-1.32) | 17.60(13.57-21.98) | 0.83(0.65-1.04) | −0.54(−0.66--0.41) |
| **Eastern Europe** | 2.55(1.87-3.42) | 0.92(0.68-1.22) | 8.42(6.16-11.40) | 2.66(1.93-3.62) | 3.58(2.92-4.25) |
| **Eastern Sub-Saharan Africa** | 1.59(1.19-2.11) | 2.28(1.68-3.04) | 3.81(2.86-5.04) | 2.40(1.79-3.19) | 0.04(−0.02-0.09) |
| **High-income Asia Pacific** | 3.10(2.44-3.82) | 1.57(1.24-1.94) | 4.55(3.41-5.71) | 0.87(0.67-1.08) | −2.09(−2.24--1.94) |
| **High-income North America** | 3.67(2.80-4.72) | 1.07(0.82-1.40) | 9.82(7.58-12.38) | 1.56(1.21-1.97) | 1.44(1.29-1.59) |
| **North Africa and Middle East** | 3.68(2.50-5.49) | 2.64(1.74-4.06) | 11.00(8.02-14.95) | 2.70(1.93-3.68) | 0.10(0.02-0.18) |
| **Oceania** | 0.03(0.02-0.04) | 0.89(0.58-1.48) | 0.06(0.04-0.09) | 0.81(0.57-1.13) | −0.52(−0.64--0.41) |
| **South Asia** | 6.43(4.55-9.11) | 1.10(0.77-1.57) | 18.67(13.61-24.92) | 1.30(0.93-1.73) | 0.51(0.45-0.56) |
| **Southeast Asia** | 4.01(2.89-5.74) | 1.62(1.15-2.40) | 11.50(8.41-15.01) | 1.86(1.35-2.40) | 0.46(0.39-0.54) |
| **Southern Latin America** | 0.82(0.57-1.12) | 1.79(1.24-2.44) | 1.48(1.07-1.99) | 1.70(1.23-2.29) | 0.26(0.13-0.39) |
| **Southern Sub-Saharan Africa** | 0.50(0.36-0.71) | 1.84(1.29-2.66) | 1.58(1.25-1.98) | 2.80(2.21-3.48) | 1.16(0.70-1.63) |
| **Tropical Latin America** | 1.13(0.84-1.49) | 1.18(0.89-1.55) | 3.57(2.68-4.59) | 1.38(1.04-1.77) | 0.76(0.66-0.87) |
| **Western Europe** | 12.18(8.87-15.75) | 2.17(1.60-2.80) | 15.69(11.84-19.57) | 1.75(1.34-2.20) | −0.75(−0.90--0.59) |
| **Western Sub-Saharan Africa** | 2.28(1.55-3.45) | 2.77(1.85-4.24) | 5.09(3.84-6.73) | 2.77(2.11-3.62) | −0.05(−0.09--0.02) |

*ASMR* Age-standardized mortality rate, *EAPC* Estimated annual percentage change, *CI* Confidence interval, *SDI* Socio-demographic index, *UI* Uncertainty interval.

## Risk factors attributable to NAFLD burden

To further identify risk factors for NAFLD, we conducted a detailed analysis of global data from 1990 to 2021. High fasting plasma glucose and Tobacco were the main risk factors for NAFLD (Fig 6, S6 Table).

Globally, the proportion of deaths attributed to high fasting plasma glucose was 4.07% in1990, and increased to 7.32% in 2021. Consistent with the proportion of deaths, the proportion of DALYs attributed to high fasting plasma glucose was 3.41% in 1990, and increased to 5.85% in 2021. Significant increases were observed to varying degrees in high SDI, high-middle SDI, middle SDI, low-middle SDI, and low SDI regions. These suggested the importance of fasting plasma glucose management for NAFLD control (Fig 6A, 6B).

**Table 4. DALYs and ASDR of NAFLD in 1990 and 2021, and temporal trends.**

| | 1990 | | 2021 | | 1990-2021 EAPC of ASDR (95%CI) |
|---|---|---|---|---|---|
| | DALYs, No.×10³ | ASDR per 100,000 (95% UI) | DALYs, No.×10³ | ASDR per 100,000 (95% UI) | |
| Global | 1689.25(1292.48-2206.53) | 40.20(30.73-52.23) | 3667.27(2903.58-4607.31) | 42.40(33.60-53.31) | 0.16(0.10-0.23) |
| **Socio-Demographic Index** | | | | | |
| Low SDI | 120.72(88.33-164.48) | 47.45(34.47-64.59) | 251.89(195.51-324.23) | 42.89(33.61-55.57) | −0.42(−0.47--0.38) |
| Low-middle SDI | 284.75(207.62-401.33) | 41.94(30.32-58.47) | 740.42(563.62-963.48) | 47.65(35.77-61.75) | 0.46(0.42-0.49) |
| Middle SDI | 491.31(375.15-633.40) | 42.14(32.23-54.23) | 1260.02(991.41-1590.49) | 45.53(36.02-56.74) | 0.25(0.21-0.29) |
| High-middle SDI | 386.00(294.44-496.60) | 37.79(28.89-48.32) | 713.36(554.27-916.76) | 38.00(29.71-48.17) | 0.02(−0.20-0.24) |
| High SDI | 404.32(313.34-522.66) | 38.39(29.45-49.78) | 697.59(548.69-876.32) | 38.73(30.51-48.83) | 0.02(−0.07-0.12) |
| **Region** | | | | | |
| Andean Latin America | 28.44(19.67-38.62) | 124.44(86.26-170.30) | 86.18(60.09-120.25) | 142.16(98.60-198.78) | 0.35(0.25-0.46) |
| Australasia | 4.96(3.78-6.48) | 21.93(16.69-28.86) | 15.51(12.41-18.96) | 33.24(26.63-40.62) | 1.63(1.49-1.77) |
| Caribbean | 20.07(14.40-26.82) | 73.84(52.88-98.81) | 42.92(30.01-59.39) | 80.31(56.31-110.84) | 0.25(−0.02-0.52) |
| Central Asia | 29.16(21.98-38.32) | 57.94(43.86-76.18) | 83.07(60.17-114.19) | 92.24(67.18-124.97) | 1.59(1.33-1.85) |
| Central Europe | 56.46(41.62-75.85) | 38.08(28.09-51.07) | 90.83(66.04-123.72) | 47.90(34.85-64.84) | 0.33(0.14-0.52) |
| Central Latin America | 110.82(79.84-148.65) | 115.68(82.48-156.37) | 362.14(270.51-471.17) | 138.84(104.37-180.55) | 0.54(0.45-0.64) |
| Central Sub-Saharan Africa | 13.06(8.51-20.01) | 47.73(31.27-74.29) | 30.51(19.96-45.91) | 43.12(28.16-65.78) | −0.44(−0.52-0.35) |
| East Asia | 263.44(202.68-330.42) | 27.43(21.28-34.39) | 435.19(336.69-552.06) | 20.11(15.61-25.11) | −0.92(−1.05--0.79) |
| Eastern Europe | 72.75(53.02-98.82) | 26.28(19.11-35.14) | 270.21(193.02-372.71) | 91.40(65.18-125.26) | 4.09(3.25-4.93) |
| Eastern Sub-Saharan Africa | 47.08(35.63-61.64) | 55.99(41.95-74.00) | 113.48(85.89-151.39) | 57.52(42.96-76.31) | −0.06(−0.12-0.00) |
| High-income Asia Pacific | 77.86(63.12-97.11) | 37.92(30.69-47.13) | 79.47(61.38-97.55) | 18.75(14.82-23.05) | −2.54(−2.69--2.38) |
| High-income North America | 98.11(74.31-129.09) | 30.13(22.32-39.78) | 241.65(188.29-307.00) | 42.07(32.96-54.17) | 1.36(1.20-1.51) |
| North Africa and Middle East | 90.35(63.41-124.27) | 53.96(37.45-78.19) | 276.31(205.33-374.13) | 58.96(43.63-79.78) | 0.36(0.32-0.40) |
| Oceania | 0.94(0.60-1.49) | 25.02(16.37-40.77) | 2.06(1.46-2.89) | 22.07(15.71-31.07) | −0.60(−0.72--0.48) |
| South Asia | 209.50(151.60-294.20) | 30.47(21.52-43.06) | 540.64(397.35-722.27) | 33.74(24.71-44.57) | 0.28(0.25-0.31) |
| Southeast Asia | 120.55(87.92-169.97) | 41.58(30.11-59.29) | 313.32(235.05-411.93) | 45.34(34.02-58.79) | 0.27(0.21-0.34) |
| Southern Latin America | 23.16(16.24-32.05) | 49.62(34.83-68.63) | 36.87(26.79-50.79) | 43.99(31.93-60.55) | 0.08(−0.06-0.22) |
| Southern Sub-Saharan Africa | 15.81(11.73-21.32) | 49.55(36.00-69.25) | 47.07(37.28-59.70) | 72.65(57.59-91.06) | 1.08(0.58-1.58) |
| Tropical Latin America | 37.57(27.87-50.07) | 34.59(25.32-46.35) | 101.28(74.62-133.35) | 38.55(28.63-50.46) | 0.53(0.42-0.64) |
| Western Europe | 304.93(225.76-399.48) | 58.35(42.69-76.44) | 349.70(269.84-441.85) | 45.31(35.03-57.09) | −0.89(−1.07--0.70) |
| Western Sub-Saharan Africa | 64.23(44.11-93.54) | 66.57(45.48-98.09) | 148.87(110.16-198.13) | 65.31(49.29-86.28) | −0.11(−0.15--0.07) |

*ASDR* Age-standardized DALYs rate, *EAPC* Estimated annual percentage change, *CI* Confidence interval, *SDI* Socio-demographic index, *UI* Uncertainty interval.

As for the other main risk factor, the proportion of deaths attributed to tobacco was 2.44% in 1990, and 2.49% in 2021. Similar with the proportion of deaths, the proportion of DALYs attributed to tobacco was 2.51% in 1990, and 2.38% in 2021. There were no clear change trends overall (Fig 6A, 6B).

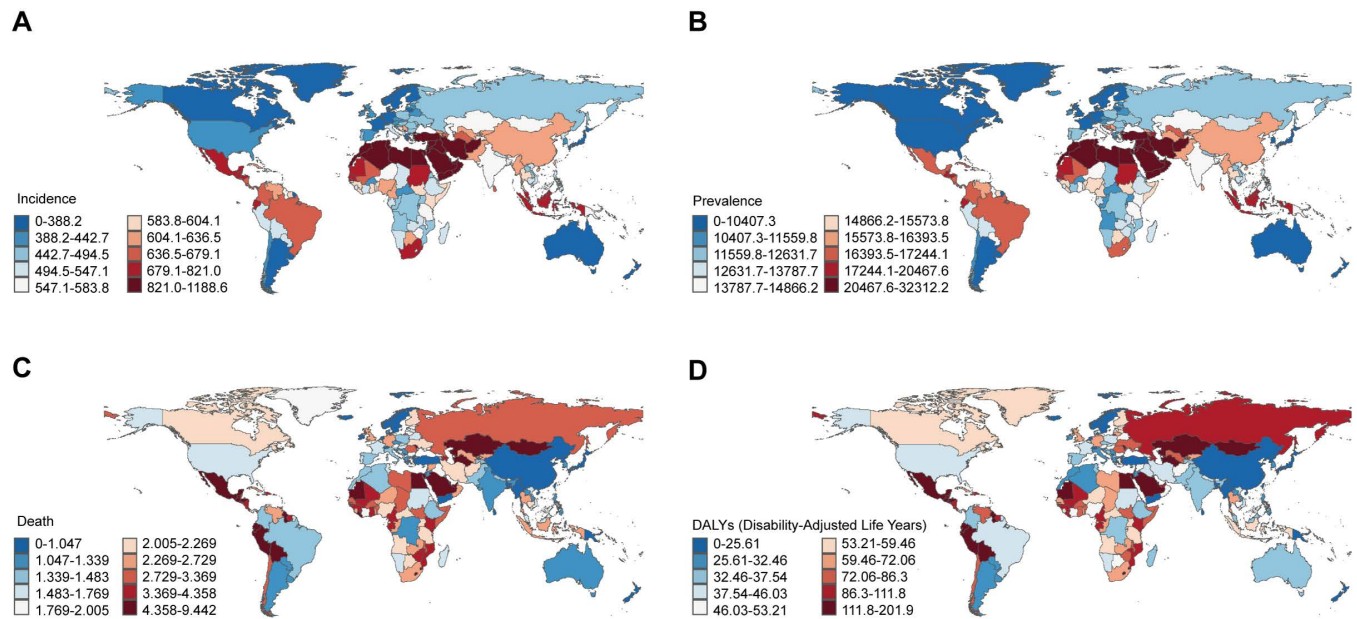

**Fig 1. Global disease burden of NAFLD for both sexes in 204 countries and territories.** The map images derive from open-source public domain data provided by Natural Earth. (A) ASIR of NAFLD in 2021; (B)ASPR of NAFLD in 2021; (C) ASMR of NAFLD in 2021; (D) ASDR of NAFLD in 2021.

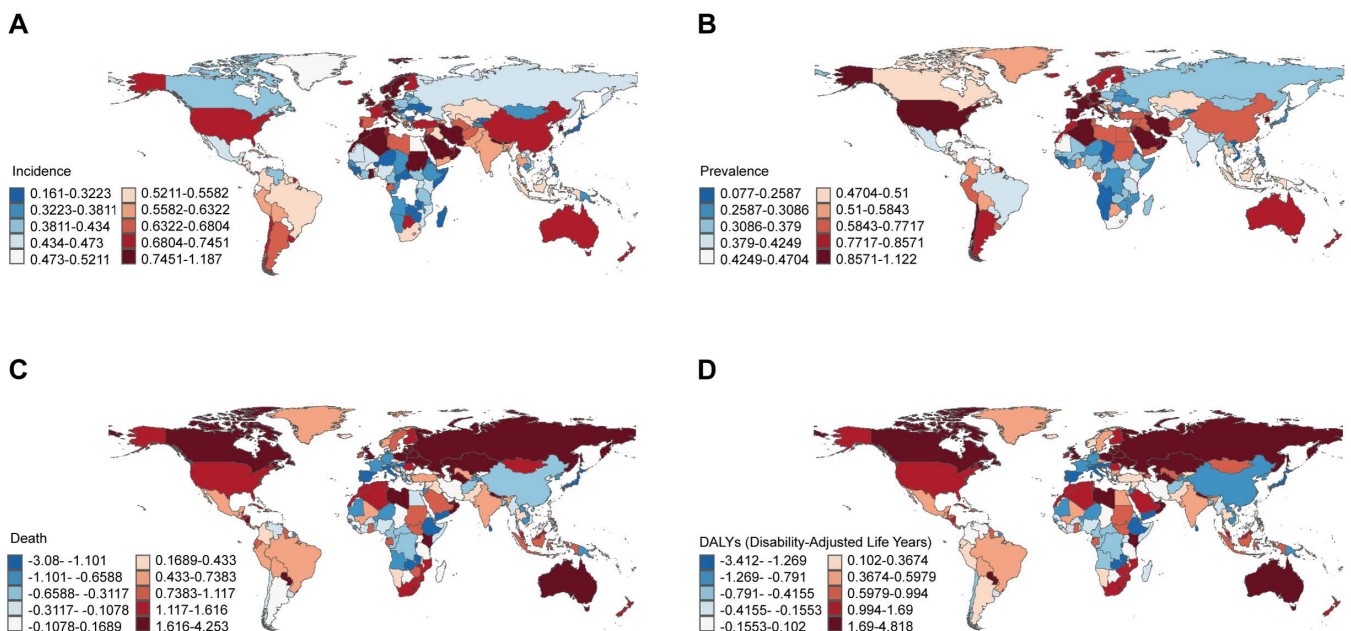

**Fig 2. Global temporal trend of NAFLD burden for both sexes in 204 countries and territories.** The map images derive from open-source public domain data provided by Natural Earth. (A)EPAC of the ASIR in NAFLD; (B)EPAC of the ASPR in NAFLD; (C) EPAC of the ASMR in NAFLD; (D) EPAC of the ASDR in NAFLD.

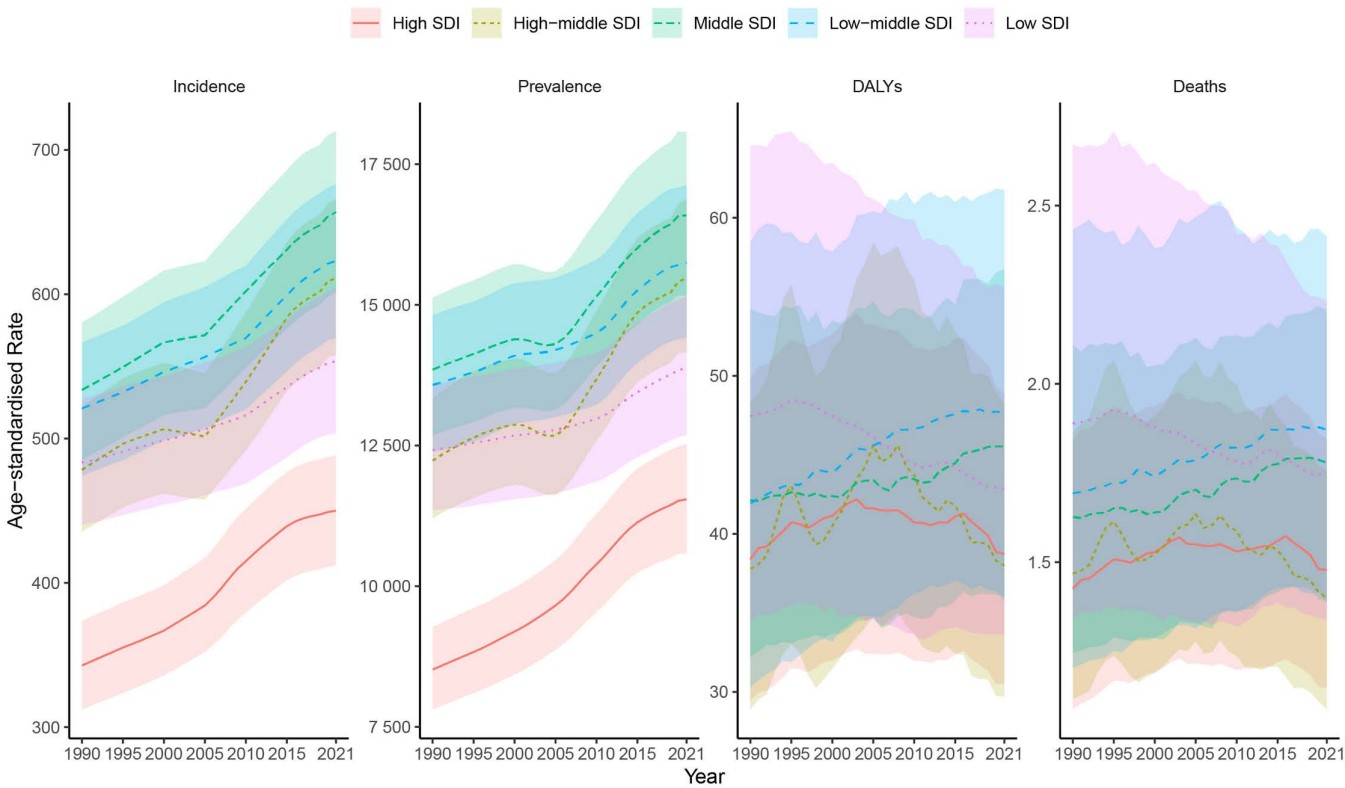

**Fig 3. Trends in Global disease burden of NAFLD at Different SDI Levels from 1990 to 2021.**

## Projection of NAFLD

To learn about the trends of NAFLD after 2021, we used Bayesian age-period-cohort models to predict the ASIR, ASPR, ASMR from 2021 to 2035 (Fig 7, S7 Table). The projections suggest that the ASIR will continue to rise consistently after 2021, from 595.12 per 100000 in 2021 to 649.41 per 100000 in 2035 (Fig 7A). Consistent with the ASIR, the ASPR will also continue to rise, from 15053.24 per 100,000 in 2021 to 16059.91 per 100,000 in 2035 (Fig 7B). The overall burden of disease will continue to increase over time. ASMR may also increase with time, from 1.63 per 100000 in 2021 to 1.67 per 100000 in 2035 (Fig 7C).

## Discussion

In this study, we leveraged publicly available modeling data and methodologies from the GBD 2021 to present the most up-to-date and comprehensive information on the incidence, prevalence, mortality, and Disability-Adjusted Life Years (DALYs) rates of NAFLD across 204 countries and territories from 1990 to 2021. Our findings contribute to estimating the global health impact of NAFLD.

Our data reveals a substantial surge in the global prevalence of NAFLD from 1990 to 2021, with an estimated 48.35 million cases in 2021, marking a 94% increase from the 24.86 million cases recorded in 1990. The Age-Standardized Incidence Rate (ASIR) of NAFLD rose by 25%, while the Age-Standardized Prevalence Rate (ASPR) increased by 24%. Furthermore, there was an upward trend in both NAFLD-related deaths and Disability-Adjusted Life Years (DALYs) between 1990 and 2021. These results underscore the continuing significant health, economic, and societal ramifications

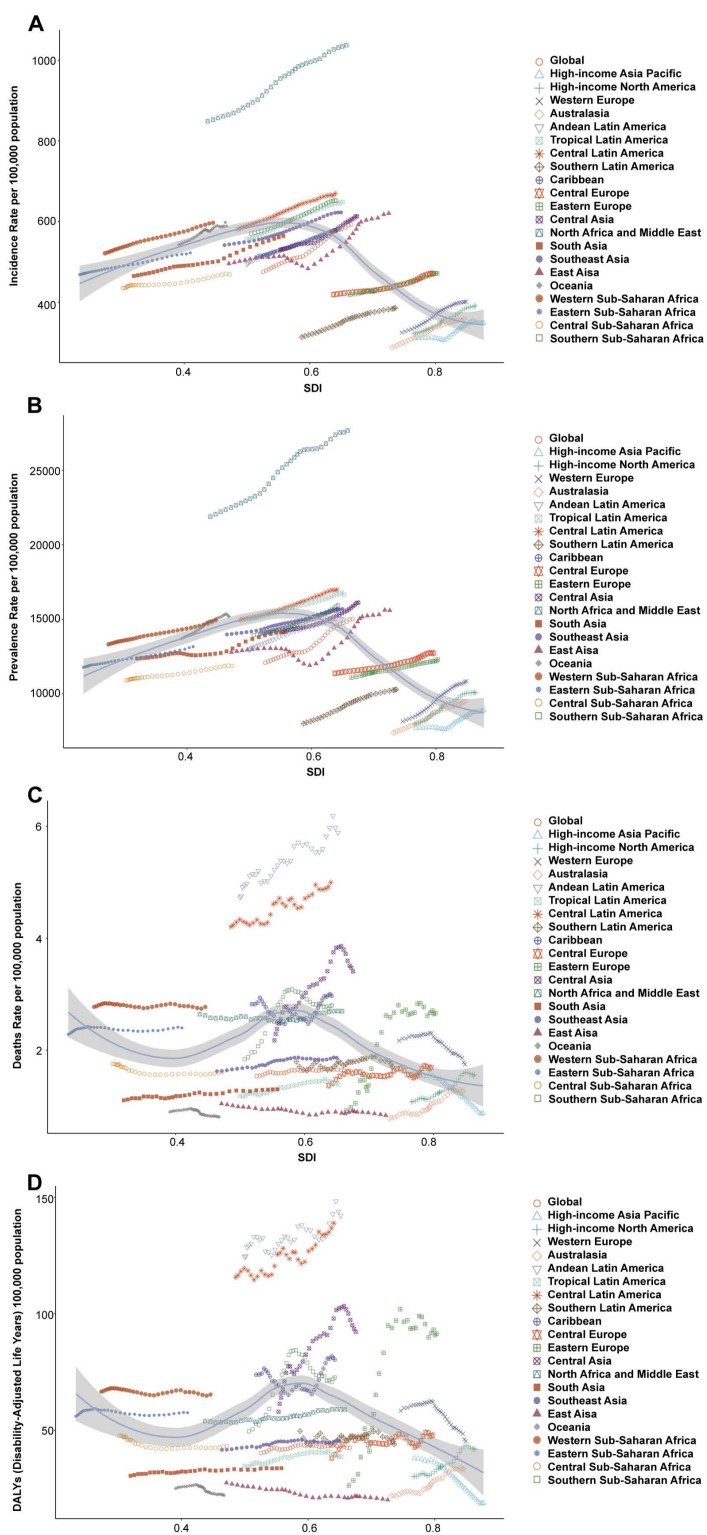

**Fig 4. Age-standardized rates of NAFLD among regions based on SDI from 1990 to 2021.** (A) ASIR; (B) ASPR; (C) ASMR; (D) ASDR.

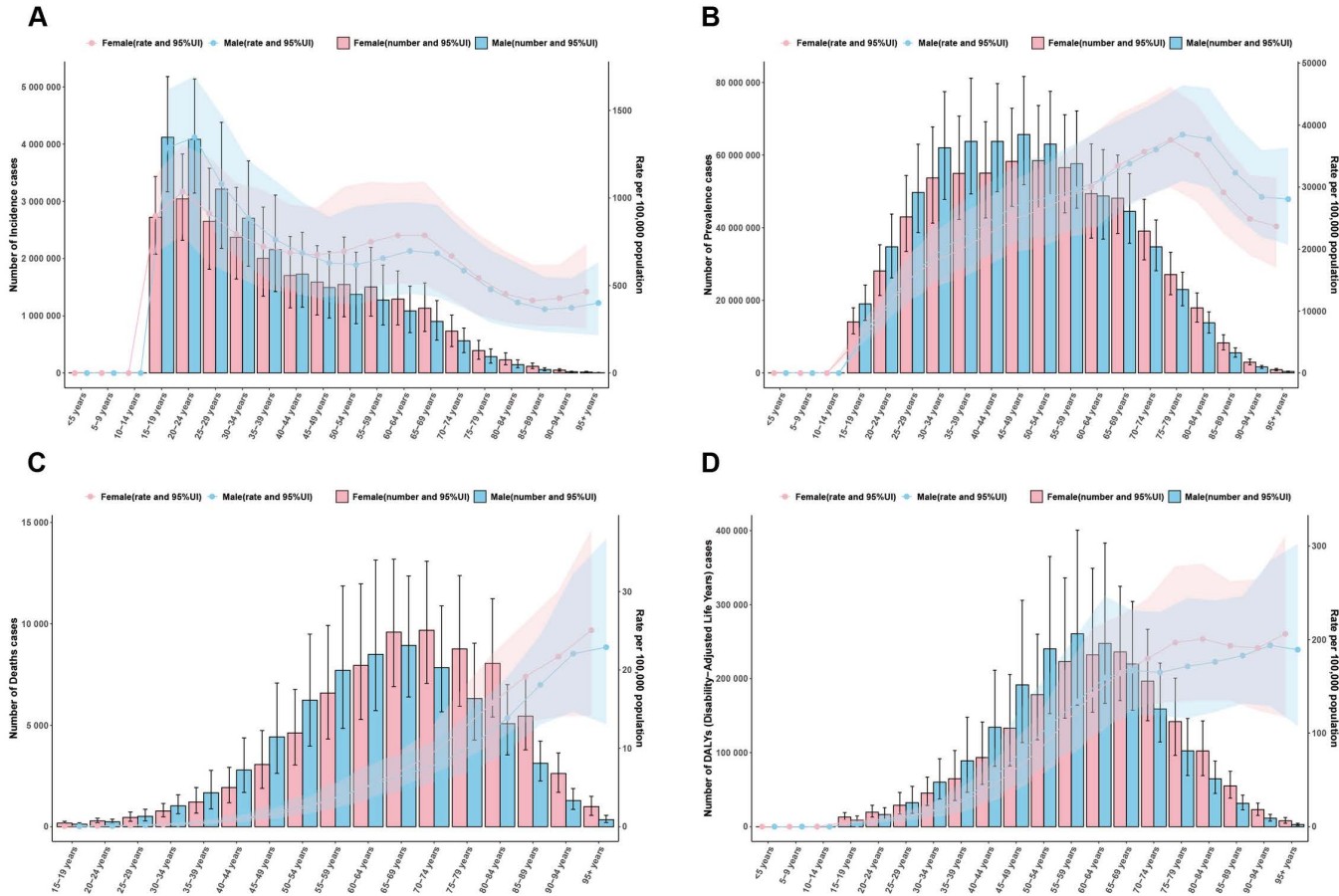

**Fig 5. Global disease burden of NAFLD in males and females by different age group in 2021.** (A) incidence cases and ASIR; (B) prevalence cases and ASPR; (C) death cases and ASMR; (D) DALYs and ASDR.

of NAFLD. With the escalating prevalence and related diseases, the healthcare, nursing expenses, and financial strain posed by NAFLD are exceptionally heavy [24]. In the United States, the estimated lifelong nursing costs for all NASH patients in 2017 amounted to approximately $222 billion [25]. Younossi et al. found that for obese patients with NASH, the cumulative direct medical costs were expected to be $1208.47 billion, while for patients with NASH but not obese, the cumulative direct medical costs were expected to be $453.88 billion [26]. By 2039, the medical costs caused by NASH per patient are expected to increase from $3636 to $6968. In Europe, the total costs related to NASH in 2018 ranged from 8.548 billion euros to 19.546 billion euros.

In this analysis, we found significant differences in the regional NAFLD prevalence rates. In 2021, the regions of North Africa and the Middle East exhibited ASIR of 1037.64 per 100,000 population and ASPR of 27686.69 per 100,000 population. Meanwhile, the Andean and Central Latin American regions reported ASMR of 5.89 and 5.00 per 100,000 population, respectively, and ASDR of 142.16 and 138.84 per 100,000 population, respectively, for the same year. The data presented in this article holds significant importance. Although NAFLD was initially considered a "Western" disease, it affects one-third of the population worldwide. In fact, NAFLD is more prevalent in regions that are still developing, such as South America, the Middle East and North Africa, Asia, and many other countries [3]. Middle SDI regions experience rapid urbanization-driven dietary shifts and sedentary lifestyles, outpacing preventive healthcare capacity, creating unchecked

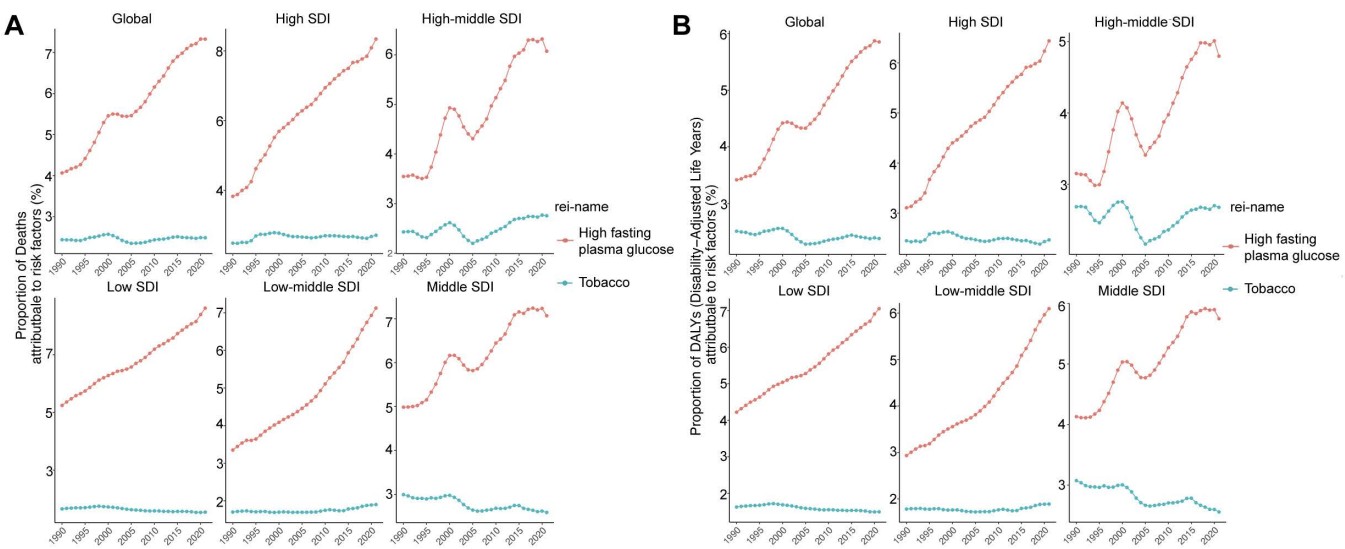

**Fig 6. Main risk factors for NAFLD from 1990 to 2021.** (A) the proportion of deaths attributed to risk factors. (B) the proportion of DALYs attributed to risk factors.

obesogenic environments. High SDI regions benefit from established public health interventions and health literacy. Nationwide NAFLD screening in countries enables early intervention. This critical regional outliers beyond socioeconomic gradients not merely from SDI gradients but distinct pathogenic landscapes. In North Africa/Middle East, genetic susceptibility interacts with rapid nutrition transitions—characterized by surging consumption of sugar-sweetened beverages and palm oil—and endemic hypovitaminosis D (>80% prevalence) [27], collectively amplifying steatosis risk independently of obesity [28,29]. The overall trend is that the rate of increase in obesity is highest in low- to lower-middle-income countries, potentially due to significant socioeconomic disparities leading to differences in diet and exercise levels. It is apparent that individuals with a lower socioeconomic status (SES) exhibit higher mortality rates in cases of NAFLD, and socially marginalized groups are at an elevated risk of contracting NAFLD and its associated complications [30]. Conversely, Latin America's mortality crisis reflects structural healthcare constraints. Delayed diagnosis occurs due to critical shortages in non – invasive diagnostics. Also, there is limited tertiary care for advanced cirrhosis and negligible transplant access. At the same time, syndemic comorbidities compound the problem [31]. In light of this correlation between SES and health, future global initiatives ought to focus on advancing these elements across the population [32]. This encompasses guaranteeing universal access to nutritious diets, education, pedestrian-friendly environments, and top-notch healthcare services as strategic measures to alleviate the NAFLD burden.

Our results indicate that in 2021, the highest number of cases and incidence rate of NAFLD were observed in the 20–24 age group, with a gradual decline as age increases. The onset age of nonalcoholic fatty liver disease (NAFLD) is decreasing. Alarmingly, an increasing number of individuals are developing NAFLD at younger ages, which means they have more time to develop severe complications, leading to an increased burden on families and society in terms of medical care. In addition, our study findings reveal that young males are more prone to developing NAFLD than their female counterparts. Similarly, research has indicated that prior to menopause, females exhibit a reduced risk of NAFLD compared to males, but this disparity diminishes post-menopause, with similar NAFLD prevalence observed between the sexes [33]. Della Torre S. highlights that women with reproductive disorders marked by altered estrogen levels, notably polycystic ovary syndrome (PCOS), have a heightened incidence of NAFLD relative to women of typical reproductive age [34]. These results further underscore the

**A**

Incidence

**B**

Prevalence

**C**

Deaths

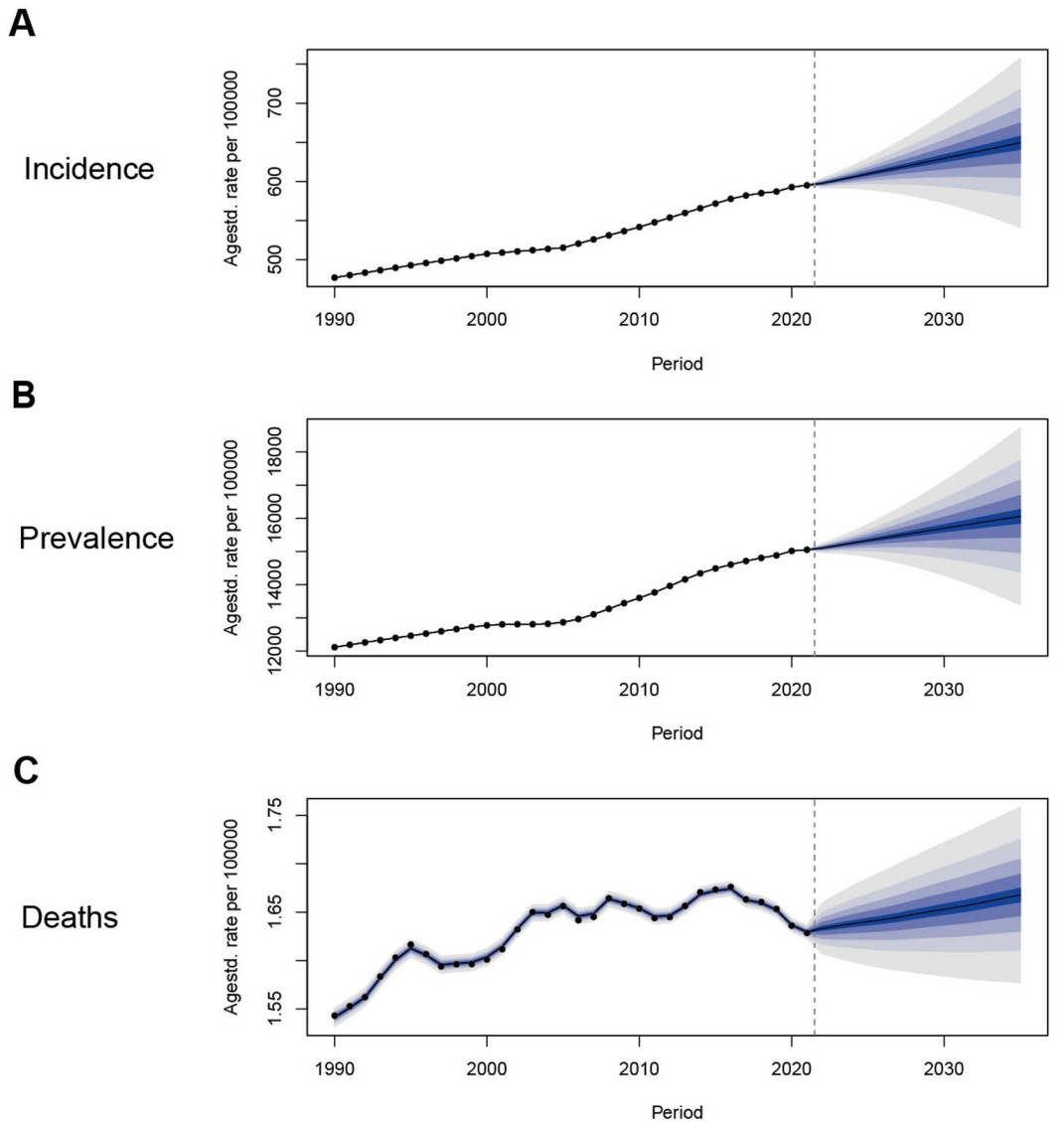

**Fig 7. Trends in the NAFLD-related ASIR, ASPR, ASMR in the Global (BAPC models): observed (dashed lines) and predicted rates (solid lines).** (A) Trends in ASIR; (B) Trends in ASPR; (C) Trends in ASMR.

protective effects of estrogen. Liver ERα signaling enables females, but not males, to mount a metabolic response to excessive dietary lipids, thereby limiting hepatic lipid accumulation in a mouse model of diet-induced NAFLD [35]. Liver ERα may serve as a sensor of hormonal and nutritional status, exerting distinct effects on hepatic metabolic regulation in both sexes, thus contributing to the gender disparity in NAFLD susceptibility [36,37].

Our study pinpoints fasting blood glucose levels and tobacco use as key risk factors for NAFLD. A study indicates that there is an independent non-linear association between fasting blood glucose and NAFLD among non-obese Chinese individuals with normal lipid levels, and an elevation in fasting blood glucose may suggest an increased risk of NAFLD [38]. Elevated fasting blood glucose levels may increase the risk of NAFLD through multiple mechanisms. Numerous studies have shown that glucose stimulates de novo lipogenesis (DNL) in the liver by activating the carbohydrate-responsive

element-binding protein (ChREBP) [39]. Furthermore, research indicates that fluctuations in glucose levels increase oxidative stress both in vivo and in vitro, leading to hepatocyte apoptosis, fibrosis, and inflammation [40]. This suggests that further exploration of the role of fasting blood glucose levels in NAFLD outcomes is of potential importance. Increasing amounts of data suggest that both passive and active smoking may be environmental stressors contributing to the progression of NAFLD [41–43], indicating that smoking may alter the regulatory role of AMP-activated kinase in lipid metabolism. Future studies should closely investigate the clinical relevance of smoking and NAFLD. However, in the meantime, smoking cessation can be considered in the management of NAFLD patients.

To gain deeper insights into the global trends of NAFLD post-2021, we employed a Bayesian Age-Period-Cohort model to precisely predict and analyze the ASIR, ASPR, and ASMR within this population cohort from 2021 to 2035. Over time, the global trend of NAFLD is projected to exhibit a sustained increase in prevalence and an escalating disease burden. Consequently, addressing this growing burden necessitates tailored interventions (such as adopting a Mediterranean diet, weight loss, exercise, and raising awareness) for different regions [44]. Furthermore, NAFLD as a global health issue underscores the need for attention from the World Health Organization (WHO) to tackle this escalating health concern worldwide.

Our study also has limitations. The first significant limitation is the underrepresentation of less developed countries in the reported epidemiological data on NAFLD. Secondly, as the research is based on the GBD 2021 database, currently only risk factor estimates at the national and regional levels are provided, and it is still impossible to conduct analyses at a finer granularity (such as provincial or city-level). Therefore, we are unable to assess the differences in risk factor exposure among different regions within a country, which to some extent limits the application of the results in local public health decision-making. Thirdly, individual SDI components—including adolescent fertility rates, population education levels, and per capita income—and NAFLD burden. Future research should disentangle how specific SDI components—particularly education gaps, income inequality, and early fertility patterns—independently modulate NAFLD epidemiology through individual-level studies, complementing our macro-level assessment. Another important limitation is the unexplained heterogeneity observed in some of the studies included in our analysis.

In summary, our study is an up-to-date assessment of the worldwide NAFLD, encompassing 204 nations, 21 regions with disease burdens, and 5 SDI areas, which had not been assessed before. Our findings reveal a 24% increase in the prevalence of NAFLD over the span of three decades. Notably, in 2021, countries with a medium SDI exhibited the highest ASIR and ASPR. Additionally, the Middle East and North Africa reported the highest NAFLD prevalence. That same year, the 20–24 age bracket accounted for the highest number of cases and incidence rates, with a subsequent decline as age advanced, indicating a shifting trend towards younger onset of NAFLD. Notably, metabolic risk factors in children and adolescents constitute one of the biggest threats to global health in the coming decades. We anticipate a steady escalation in the global NAFLD burden by 2035, imposing substantial societal strain. As NAFLD is reversible, initiating public health initiatives to enhance awareness, improve diagnostic accuracy, and encourage dietary and exercise interventions can mitigate the projected surge in disease burden.

## Supporting information

**S1 Table. ASIR, ASPR, ASMR, ASDR of NAFLD for both sexes in 204 countries and territories in 2021.**
(XLSX)

**S2 Table. EAPC of the ASIR, ASPR, ASMR, ASDR for both sexes in NAFLD in 204 countries and territories from 1990–2021.**
(XLSX)

**S3 Table. ASIR, ASPR, ASMR, ASDR trends in global disease burden of NAFLD at different SDI levels from 1990 to 2021.**
(XLSX)

**S4 Table. ASIR, ASPR, ASMR, ASDR of NAFLD among regions based on SDI from 1990 to 2021.**
(XLSX)

**S5 Table. Global disease burden of NAFLD in males and females by different age group in 2021.** Incidence cases and ASIR; prevalence cases and ASPR; death cases and ASMR; DALYs and ASDR.
(XLSX)

**S6 Table. The proportion of deaths and DALYs attributed to risk factors of NAFLD from 1990 to 2021.**
(XLSX)

**S7 Table. Trends in the NAFLD-related ASIR, ASPR, ASMR in the global (BAPC models).**
(XLSX)

## Author contributions

**Data curation:** Minxiu Wang, Yuqin Mao, Shaoyan Xuan.

**Formal analysis:** Jiqing Du, Jiong Wang, Shaoyan Xuan.

**Investigation:** Jiqing Du, Jiong Wang, Shu Yang.

**Resources:** Jiong Wang, Zhihua Tang.

**Software:** Shaoyan Xuan.

**Writing – original draft:** Minxiu Wang.

**Writing – review & editing:** Yuqin Mao, Baoguo Li, Zhihua Tang.

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
