## [Decision Letter · Decision Letter 0]

27 May 2025

Dear Dr. Wang,

We look forward to receiving your revised manuscript.

Kind regards,

Hamidreza Karimi-Sari

Academic Editor

PLOS ONE

Journal Requirements:

2. We note that your Data Availability Statement is currently as follows: [All relevant data are within the manuscript and its Supporting Information files]

3. We note that Figures 1 & 2 your submission contain [map/satellite] images which may be copyrighted. All PLOS content is published under the Creative Commons Attribution License (CC BY 4.0), which means that the manuscript, images, and Supporting Information files will be freely available online, and any third party is permitted to access, download, copy, distribute, and use these materials in any way, even commercially, with proper attribution. For these reasons, we cannot publish previously copyrighted maps or satellite images created using proprietary data, such as Google software (Google Maps, Street View, and Earth). For more information, see our copyright guidelines: http://journals.plos.org/plosone/s/licenses-and-copyright.

a. You may seek permission from the original copyright holder of Figures 1 & 2 to publish the content specifically under the CC BY 4.0 license. 

Reviewers' comments:

Reviewer's Responses to Questions

**Comments to the Author**

1. Is the manuscript technically sound, and do the data support the conclusions?

Reviewer #1: Partly

2. Has the statistical analysis been performed appropriately and rigorously?

Reviewer #1: Yes

3. Have the authors made all data underlying the findings in their manuscript fully available?

Reviewer #1: Yes

4. Is the manuscript presented in an intelligible fashion and written in standard English?

Reviewer #1: Yes

Reviewer #1: The authors assessed the global burden of NAFLD from 1990 to 2021 encompassing 204 countries, 21 regions with varying disease burdens, and 5 SDI areas. Projections of global burden trends were made using a Bayesian age-period-cohort (BAPC) model. The study provides informative summaries of increasing NAFLD burden overall and national level. My questions below are related to projection, estimation of association between risk factors and NAFLD burden, and their interpretations which needs further clarification.

The authors used BAPC models to estimate reasonable predictions of global burden trends. Additional details would help understand the approach. What prior was used? Has sensitivity analysis done based on the different choices of priors? The authors also mention the prediction yields precise predictions of ASIR, ASPR, and ASMR (line 287). Please elaborate on the reliability and robustness of the predictions.

From Figure 3, high SDI group has the lowest incidence and prevalence of NAFLD, but all other groups have wide overlap. How is the uncertainty represented by the ribbons around the estimates captured here? Are these model-based confidence intervals? If not, have the authors considered fitting a model to test whether middle SDI has the highest and high SDI has the lowest ASIR and ASPR trends over time?

Why is there such a big gap of ASIR and ASPR trends between high SDI vs. other SDI? Do the authors have hypothesis driving this?

The authors used SDI calculated from several social factors, including the fertility rate of the population aged <25 years, the education level of the population aged >15 years, and per capita income. Have the authors investigated the association of these individual factors to NAFLD burden?

The risk factors were assessed at country and regional levels. Were data available on a more granular level? If not, would the authors see this as a limitation to the study?

The inverted U-shaped relationship depicted in the figures seems to be heavily influenced by the outlier regions. I suggest removing the smoothed lines unless they are model based. Can the authors elaborate on the effects beyond SDI that may influence these outliers (North Africa and Middle East has the highest prevalence and incidence and Central and Andean Latin America have highest death rate and DALYs)?

Minor points

Abstract Results line 32 & Results line 129: it is unclear what “this age group” refers to.

Abstract Results line 35: SDI needs to be defined.

The fonts of texts and legends in figures 4 and 5 should be increased.

**Do you want your identity to be public for this peer review?** For information about this choice, including consent withdrawal, please see our Privacy Policy

Reviewer #1: No

---

## [Author Response · Author response to Decision Letter 1]

9 Jul 2025

Response to reviewers’ comments

Journal Requirements: #1-1: Please ensure that your manuscript meets PLOS ONE's style requirements, including those for file naming. The PLOS ONE style templates can be found at

Response: Thanks. We confirm that we have carefully reviewed and strictly adhered to all PLOS ONE style guidelines as outlined in the provided templates and author instructions. Should you require any additional clarification or specific file verification, please do not hesitate to let us know.

Journal Requirements: #1-2: We note that your Data Availability Statement is currently as follows: [All relevant data are within the manuscript and its Supporting Information files]

Response: Thank you for the reminder. In this revision, we have included all the raw data required to replicate the results of the study. The relevant information has been added to the main text and supplementary materials.

Journal Requirements: #1-3: We note that Figures 1 & 2 your submission contain [map/satellite] images which may be copyrighted. All PLOS content is published under the Creative Commons Attribution License (CC BY 4.0), which means that the manuscript, images, and Supporting Information files will be freely available online, and any third party is permitted to access, download, copy, distribute, and use these materials in any way, even commercially, with proper attribution. For these reasons, we cannot publish previously copyrighted maps or satellite images created using proprietary data, such as Google software (Google Maps, Street View, and Earth).

Response: Response: Thank you for raising questions regarding the map images in Figure 1 and 2. Please find our responses below:

a) Source of Map Images:

The map images in Figure 1 and 2 were created using the rnaturalearth R package, which provides access to Natural Earth map data. This data is open-source and available at https://github.com/ropensci/rnaturalearth.

b) Copyright Status:

To our knowledge, the map data from Natural Earth provided through the rnaturalearth package is in the public domain and therefore not subject to copyright restrictions. Natural Earth map data is widely used for world mapping and is explicitly designed for open, unrestricted use.

c) Permission for Use:

Since the map images derive from open-source public domain data provided by Natural Earth, we believe no additional copyright permission is required. However, if necessary, we can include a statement attributing Natural Earth as the data source.

Reviewer #1: The authors assessed the global burden of NAFLD from 1990 to 2021 encompassing 204 countries, 21 regions with varying disease burdens, and 5 SDI areas. Projections of global burden trends were made using a Bayesian age-period-cohort (BAPC) model. The study provides informative summaries of increasing NAFLD burden overall and national level. My questions below are related to projection, estimation of association between risk factors and NAFLD burden, and their interpretations which needs further clarification.

Response: We sincerely thank for their thoughtful evaluation of our work and their recognition of the study's global scope and methodological approach.

Reviewer #1-1: The authors used BAPC models to estimate reasonable predictions of global burden trends. Additional details would help understand the approach. What prior was used? Has sensitivity analysis done based on the different choices of priors? The authors also mention the prediction yields precise predictions of ASIR, ASPR, and ASMR (line 287). Please elaborate on the reliability and robustness of the predictions.

Response: Thank you for your valuable suggestion. Because of the relatively lower absolute percentage deviation for the BAPC model, we used it for the projections of NAFLD incidence rates and case numbers through 2035.

The APC model assumes there is a multiplicative effect of age, period and cohort,

Y_ap=μ'〖α'〗_a 〖β'〗_p 〖γ'〗_c,

Where Y_ap denotes the incident case counts, 〖α'〗_adenotes the age effect, 〖β'〗_p denotes the period effect and〖 γ'〗_cdenotes the cohort effect. Where μ, α_a, β_p and γ_c are the logarithms of μ', 〖α'〗_a, 〖β'〗_p and 〖γ'〗_c, respectively.

We conducted a BAPC analysis with integrated nested Laplace approximation (INLA). To ensure smoothing, BAPC models assume independent mean-zero normal distributions on the second differences of all effects. Specifically, the BAPC model assumes prior distribution of the age effect as follows:

f(α│k_α )∝k_α^((t-2)/2) exp⁡{-k_α/2 ∑_(i=3)^I▒[(α_i-α_(i-1) )-(α_(i-1)-α_(i-2) )]^2 },

Considering that we are interested in the incident case counts for age group a, with a t period into the future, the following equation can be applied:

〖log⁡(Y〗_(a,p+t))=〖μ+α〗_a+β_(p+t)+γ_(c+t)+δ_(a,p+t),

Here, we add an independent random effect δ_(a,p+t)∼N(0,k_δ^(-1) ) to adjust for overdispersion. Considering the smoothing assumption, the BAPC models assume prior distribution of the period effect as follows:

β_(p+t) |β_1 〖,…,β〗_p,k_β∼N((1+t)β_p-tβ_(p-1),k_β^(-1) (1+2^2+⋯+t^2)),

The summary estimates (mean, standard deviation, 2.5% quantile, median and 97.5% quantile) of all variance parameters in the BAPC models can be obtained. (PMID: 33349860) To evaluate the reliability and robustness of these predictions, we compared the mean values and 95% UI predicted by the BAPC model from 1990 to 2021 with the actual values. The predicted values were within the 95% UI of the actual values, thereby confirming the reliability and robustness of the model. For example, in 1990, the ASIR in the global was 477.04 (95% UI: 476.85–477.23) predicted by BAPC models (S7 Table), and the actual value, it was 475.54 (95% UI: 432.59-518.19) (Table 1).

Reviewer #1-2: From Figure 3, high SDI group has the lowest incidence and prevalence of NAFLD, but all other groups have wide overlap. How is the uncertainty represented by the ribbons around the estimates captured here? Are these model-based confidence intervals? If not, have the authors considered fitting a model to test whether middle SDI has the highest and high SDI has the lowest ASIR and ASPR trends over time?

Response: Thank you for your valuable suggestion. The ribbons in Figure 3 represent the 95% uncertainty intervals (UIs), which are based on DisMod model used in the GBD study. It is used to reflect the estimated values of incidence or DALYs of SDI in different regions in a certain year, as well as the uncertain intervals. For example, in 1990, the ASIR for the middle SDI group was 534 (95% UI: 485–581), and for the low-middle SDI group, it was 521 (95% UI: 474–568). Although the ASIR of the middle SDI group is estimated to be higher than that of the low-middle SDI group, due to the overlap of the 95% UI, the difference cannot be considered statistically significant. In addition, regarding the trend of SDI groups changing over the years, we have analyzed it through the estimated annual percentage change (EAPC). The results are shown in Table 1-4 to present the changing trends of different SDI groups during the study period.

Reviewer #1-3: Why is there such a big gap of ASIR and ASPR trends between high SDI vs. other SDI? Do the authors have hypothesis driving this?

Response: Thanks for the question. We propose the inverted U-shaped relationship between SDI and NAFLD burden (peak in middle SDI, lowest in high SDI) arises from intersecting epidemiological transitions and healthcare disparities: 1) Middle SDI regions experience rapid urbanization-driven dietary shifts and sedentary lifestyles, outpacing preventive healthcare capacity, creating unchecked obesogenic environments; 2) Importantly, the incidence and prevalence of NAFLD are relatively low in high SDI regions, which can be attributed to the effective public health measures and the population's higher health awareness. Nationwide NAFLD screening in countries enables early intervention. Furthermore, high SDI regions often have policies and regulations in place that promote healthier environments. This can include initiatives to reduce the availability and consumption of unhealthy foods, as well as promoting physical activity through urban planning and public health campaigns. 3) The relatively high EAPC in high-SDI regions paradoxically reflects their advanced healthcare capacity: expanded non-invasive screening captures subclinical NAFLD cases that would remain undetected in resource-limited settings, while aging populations manifest prevalence peaks (75-79y). Residual heterogeneity may reflect unmeasured genetic/environmental factors, warranting future granular studies.(PMID: 34668658, 39980749)

Reviewer #1-4: The authors used SDI calculated from several social factors, including the fertility rate of the population aged <25 years, the education level of the population aged >15 years, and per capita income. Have the authors investigated the association of these individual factors to NAFLD burden?

Response: Thanks for raising this important point. We thank the reviewer for raising this important point regarding the potential independent associations between individual SDI components—including adolescent fertility rates, population education levels, and per capita income—and NAFLD burden. While our current study utilized the composite SDI metric as an integrated indicator of socioeconomic development (following GBD methodology), we acknowledge that examining these constituent factors separately could yield deeper insights. Emerging evidence suggests distinct pathways: lower educational attainment may constrain health literacy and preventive behaviors; income disparities likely create differential access to nutritious diets and healthcare; while high adolescent fertility could induce metabolic stressors during critical developmental periods. Although our aggregated approach precluded granular analysis of these relationships, we recognize their significance and will address this limitation in the revised Discussion.

Reviewer #1-5: The risk factors were assessed at country and regional levels. Were data available on a more granular level? If not, would the authors see this as a limitation to the study?

Response: Thank you for your valuable suggestion. This study relies on the GBD 2021 dataset, risk factors cannot obtain data on a more granular level. We acknowledge this as a limitation of our study and have explained it in the discussion section ‘Secondly, as the research is based on the GBD 2021 database, currently only risk factor estimates at the national and regional levels are provided, and it is still impossible to conduct analyses at a finer granularity (such as provincial or city-level). Therefore, we are unable to assess the differences in risk factor exposure among different regions within a country, which to some extent limits the application of the results in local public health decision-making’.

Reviewer #1-6: The inverted U-shaped relationship depicted in the figures seems to be heavily influenced by the outlier regions. I suggest removing the smoothed lines unless they are model based. Can the authors elaborate on the effects beyond SDI that may influence these outliers (North Africa and Middle East has the highest prevalence and incidence and Central and Andean Latin America have highest death rate and DALYs)?

Response: Thank you for your valuable suggestion. In response, we have re-analyzed the data after removing the identified outlier regions and re-plotted the figure as follows. The results show that the inverted U-shaped relationship remains evident, suggesting that this trend is robust and not solely driven by outliers.

We attribute the exceptional NAFLD burden in North Africa/Middle East (ASPR: 27,686.69/100,000; ASIR: 1,037.64/100,000) and high mortality in Central/Andean Latin America (ASMR: 5.89/100,000; ASDR: 142.16/100,000) to region-specific amplifiers beyond SDI. In North Africa/Middle East, genetic susceptibility synergizes with rapid dietary transitions toward sugar-sweetened beverages and saturated fats, while endemic vitamin D deficiency (>80% prevalence) exacerbates insulin resistance (PMID: 26186591, 24452044). Conversely, Latin America's mortality burden stems from healthcare fragmentation—delayed diagnosis due to limited advanced diagnostics (e.g., FibroScan scarcity) and restricted transplant access—compounded by competing social risks (e.g., violence-related trauma) and dual epidemics of NAFLD with chronic viral hepatitis/H. pylori infections that accelerate liver damage (PMID: 36990226). These factors create distinct pathogenic landscapes where genetic-epigenetic interactions amplify incidence in one region, and systemic healthcare gaps convert prevalence to mortality in another. We incorporated this point into the Discussion section of the revised manuscript.

Reviewer #1-7: Abstract Results line 32 & Results line 129: it is unclear what “this age group” refers to.

Response: Thank you for pointing out the ambiguity. “This age group” refers to the all-age population. We have revised in the manuscript.

Reviewer #1-8: Abstract Results line 35: SDI needs to be defined.

Response: Thank you for your valuable suggestion. We have defined SDI to Socio-demographic index in Abstract Results line 35.

Reviewer #1-9: The fonts of texts and legends in figures 4 and 5 should be increased.

Response: Thank you for your valuable suggestion. We have revised figure 4 and 5 accordingly based on your comment.

---

## [Decision Letter · Decision Letter 1]

4 Aug 2025

Global burden of NAFLD 1990-2021 and projections to 2035: results from the global burden of disease study 2021

PONE-D-24-57511R1

Dear Dr. Wang,

We’re pleased to inform you that your manuscript has been judged scientifically suitable for publication and will be formally accepted for publication once it meets all outstanding technical requirements.

Kind regards,

Hamidreza Karimi-Sari

Academic Editor

PLOS ONE

Additional Editor Comments (optional):

Reviewers' comments:

Reviewer's Responses to Questions

**Comments to the Author**

Reviewer #1: All comments have been addressed

Reviewer #2: (No Response)

2. Is the manuscript technically sound, and do the data support the conclusions?

Reviewer #1: Yes

Reviewer #2: Yes

3. Has the statistical analysis been performed appropriately and rigorously?

Reviewer #1: Yes

Reviewer #2: Yes

4. Have the authors made all data underlying the findings in their manuscript fully available?

Reviewer #1: Yes

Reviewer #2: Yes

5. Is the manuscript presented in an intelligible fashion and written in standard English?

Reviewer #1: Yes

Reviewer #2: Yes

Reviewer #1: I would like to thank the authors for thoroughly addressing all my comments. I believe the manuscript has greater clarity and have no further questions.

Reviewer #2: Thank you to the authors for their thorough responses to the feedback and efforts to address them. I commend the authors for this rigorous, impactful and novel research. Its findings have important implications in addressing NAFLD worldwide. I just have minor comments:

Abstract: The abstract was well-written. I have a few suggestions: 1) Add predicting the projection of NAFLD in the aims sentence (“This study used...”). 2) Remove repetition in the Background and Methods. For example, the time period was repeated three times and the data source twice. 3) Briefly introduce the socio-demographic index and its source.

Figures/Tables: As mentioned by a previous reviewer, it remains unclear what the ribbons in Figure 3 represent based on the figure alone. It could be informative to add 95% UI and 95% CI in the legendary or a footnote. Similarly, Tables 1 and 2 specify either 95% UI or 95% CI for ASPR and EAPC, but not incident or prevalent cases. What do the intervals for these mean? I couldn’t find it in the methods.

Results: There was a tendency in some subsections, such as Age and sex patterns and Risk factors attributable to NAFLD burden, to interpret the results in the Results rather than the Discussion. For example, the authors wrote in lines 194-196: “...which may mean that younger men are more likely to develop NAFLD.” Additionally, in lines 213-214: “These suggested the importance of fasting plasma glucose management for NAFLD control (Fig 6A, 6B).” The Results section should report findings objectively, while interpretation of findings be reported in the Discussion.

Overall, I commend the authors for their wonderful work. I look forward to seeing the research through.

**Do you want your identity to be public for this peer review?** For information about this choice, including consent withdrawal, please see our Privacy Policy

Reviewer #1: No

Reviewer #2: No

---

## [Editor Report · Acceptance letter]

PONE-D-24-57511R1

PLOS ONE

Dear Dr. Wang,

I'm pleased to inform you that your manuscript has been deemed suitable for publication in PLOS ONE. Congratulations! Your manuscript is now being handed over to our production team.

Kind regards,

on behalf of

Hamidreza Karimi-Sari

Academic Editor

PLOS ONE